# Exploring Fourier-Transform Infrared Microscopy for Scabies Mite Detection in Human Tissue Sections: A Preliminary Technical Feasibility Study

**DOI:** 10.3390/ijms262311597

**Published:** 2025-11-29

**Authors:** Maximilian Lammer, Matthias Schmuth, Paul Bellmann, Verena Moosbrugger-Martinz, Bernhard Zelger, Birgit Moser, Roland Stalder, Christian Wolfgang Huck, Miranda Klosterhuber, Johannes Dominikus Pallua

**Affiliations:** 1Department of Dermatology, Venereology and Allergy, Medical University Innsbruck, 6020 Innsbruck, Austria; maximilian.lammer@i-med.ac.at (M.L.); matthias.schmuth@i-med.ac.at (M.S.); paul.bellmann@i-med.ac.at (P.B.); verena.martinz@i-med.ac.at (V.M.-M.); birgit.moser@tirol-kliniken.at (B.M.); 2Private Dermatological Practice, Specialized in Dermatopathology, Mariahilfpark 1/6, 6020 Innsbruck, Austria; bernhard.zelger@gmail.com; 3Institute of Mineralogy and Petrography, University of Innsbruck, 6020 Innsbruck, Austria; roland.stalder@uibk.ac.at; 4Institute of Analytical Chemistry and Radiochemistry, University of Innsbruck, 6020 Innsbruck, Austria; christian.w.huck@uibk.ac.at; 5Department of Orthopaedics and Traumatology, Medical University Innsbruck, 6020 Innsbruck, Austria; miranda.klosterhuber@i-med.ac.at

**Keywords:** infrared imaging, scabies, chitin, parasitic skin disease, formalin-fixed paraffin-embedded (FFPE) tissue, digital pathology, spectral imaging, chemometric analysis

## Abstract

Scabies, caused by *Sarcoptes scabiei* var. *hominis*, remains difficult to diagnose in histological routine when mite fragments are sparse or degraded. We explored whether Fourier-transform infrared (FTIR) microscopy can detect chitin-associated spectral signatures of scabies mites in formalin-fixed paraffin-embedded (FFPE) human skin sections and distinguish them from surrounding host tissue. FFPE sections from six patients with histologically confirmed crusted scabies were analysed by FTIR imaging, univariate mapping of selected bands, and multivariate image analysis within the 1000–1200 cm^−1^ carbohydrate region. Spectra from mite exoskeleton, stratum corneum, and dermis were compared, and absorbance at 1072 cm^−1^ was quantified across all samples. Mite regions showed consistently higher 1072 cm^−1^ absorbance than adjacent epidermal and dermal compartments, and unsupervised clustering reproducibly delineated mite-associated domains that co-localised with structures identified on haematoxylin-and-eosin-stained sections. Within this small, preliminary proof-of-concept cohort and in the absence of healthy or disease controls, the data do not allow estimation of clinical diagnostic performance at the patient level, but demonstrate the technical feasibility and analytical robustness of FTIR microscopy for intra-lesional detection of chitin-rich parasite structures in scabies lesions and provide a framework for future comparative studies in larger, prospectively collected cohorts.

## 1. Introduction

Scabies is a contagious parasitic skin infestation caused by the mite *Sarcoptes scabiei* var. *hominis*. It remains a prevalent condition globally, with hundreds of millions of cases estimated yearly [1]. In practice, scabies is often diagnosed clinically—patients typically present with intense pruritus and characteristic rash distributions along with a history of contagion. Scabies may clinically resemble other pruritic dermatoses; in the absence of universally pathognomonic features, clinical diagnosis may be challenging [2,3]. The standard diagnostic method for scabies has long been the identification of mites, eggs, or faecal pellets (scybala) via microscopic examination of skin scrapings. While microscopic skin scraping examination is a specific confirmatory test, its low sensitivity and labour-intensive nature significantly limit its utility in routine practice [4,5]. Histological examination, particularly through skin biopsy, is a diagnostic tool for identifying scabies mites in cases of scabies. While a biopsy may yield evidence of mites, eggs, or faecal material, it is considered invasive and unnecessary in typical presentations [6]. Despite the global prevalence of scabies, its diagnosis remains challenging due to the absence of a single, reliable, and non-invasive diagnostic gold standard. Common bedside methods, such as the burrow ink and adhesive tape tests, are rapid and straightforward but suffer from variable sensitivity and operator dependence [7]. Dermoscopy offers improved sensitivity and allows for direct, non-invasive visualisation of the mite and its burrow; however, it requires training and may yield false negatives in cases with low mite burden [8]. Advanced optical imaging techniques, such as video dermatoscopy, reflectance confocal microscopy, and optical coherence tomography, remain primarily restricted to research [4]. Consequently, current diagnostic tools exhibit limitations in sensitivity, specificity, accessibility, or practicality. This underscores the urgent need for innovative, patient-friendly, and accurate diagnostic approaches to overcome conventional methods and enable reliable scabies detection across diverse clinical settings.

The lack of dependable methods for quickly diagnosing scabies has generated significant interest in developing techniques such as infrared (IR) microscopy [9,10,11,12]. In the clinical context, we use the term “non-invasive” to refer to bedside techniques applied directly to the patient without tissue removal (e.g., dermoscopy, video dermoscopy, reflectance confocal microscopy, optical coherence tomography), which aim to visualise mites or burrows in vivo. By contrast, IR microscopy and, specifically, Fourier-transform infrared (FTIR) microscopy as used in this study are applied ex vivo to formalin-fixed paraffin-embedded (FFPE) biopsy sections and therefore require an invasive sampling step at the patient level. The non-destructive, label-free nature of IR microscopy at the tissue level, which helps maintain the integrity of **excised samples**, enhances its attractiveness [13]. This technique assesses the absorption and transmission of infrared radiation through various chemical bonds, providing distinct molecular insights into the sample [14]. IR microscopy presents a promising tool for analysing skin tissue, offering the potential to enhance diagnostic precision in dermatological disorders, such as scabies. This label-free technique enables the molecular characterisation of skin samples without the need for additional staining or extensive sample preparation [15]. In the context of scabies, IR microscopy may aid in detecting changes associated with mite infestation, such as alterations in the stratum corneum composition or inflammation-induced spectral signatures [16]. Its applications in skin research underscore its diagnostic versatility [17,18,19]. In the present work, FTIR microscopy should therefore be regarded as an adjunct to conventional histopathology on biopsy material rather than as a non-invasive in vivo diagnostic test. Against this background, we conducted a preliminary technical feasibility (proof-of-concept) study using FTIR microscopy on FFPE sections from six histologically confirmed scabies cases to explore whether chitin-associated spectral signatures and multivariate image analysis can delineate mite structures from surrounding host tissue in situ. Our goal was to establish analytical feasibility and robustness of the spectral marker at 1072 cm^−1^ rather than to provide validated diagnostic performance metrics.

## 2. Results

Our study group comprised six patients (three men, three women) with a mean age of 61.2 years (range 21–88 years), all of whom were diagnosed with scabies based on clinical presentation and histological confirmation. Table 1 summarises the clinical and pathological characteristics of the cohort.

Histological analysis revealed the presence of *Sarcoptes scabiei* mites or their remnants, such as chitinous exoskeletons, in all samples, often accompanied by scybala (faecal pellets). In addition, reactive hyperkeratosis with a perivascular inflammatory cell infiltrate consisting of lymphocytes, macrophages, and interstitially abundant eosinophilic granulocytes was observed in the majority of cases (Figure 1).

### 2.1. FTIR Spectral Comparison of Skin Layers and Scabies Mite Structures

FTIR microscopy enabled the molecular characterisation of different skin compartments, revealing distinct biochemical profiles for the dermis, stratum corneum, and the scabies mite exoskeleton (Figure 2).

In the dermis, the prominent Amide peaks highlight the protein content [20,21], while the stratum corneum’s spectral profile is linked to long-chain fatty acids and ceramides, essential for the skin’s barrier function [21,22,23]. The exoskeleton exhibited unique spectral features distinct from those of the host tissue, firm peaks between 1000 and 1200 cm^−1^, which are related to the presence of chitin. The mite’s spectral complexity in the low wavenumber region (<1200 cm^−1^) differentiates it from host tissues, with chitin peaks serving as a key biochemical marker [24,25,26]. A summary of the observed marker bands and their relative intensities across the dermis, stratum corneum, and scabies exoskeleton is provided in Table 2.

Collectively, these findings underscore the diagnostic value of FTIR imaging for differentiating parasitic from host structures in situ. The presence of chitin-associated spectral features provides a unique marker for detecting mite fragments within FFPE tissue sections.

### 2.2. Multivariate Image Analysis (Mia) of the 1000–1200 cm^−1^ Fingerprint Region

To enhance tissue differentiation, we applied MIA to FTIR data in the 1000–1200 cm^−1^ fingerprint region. Based on the preprocessing pipeline described in Section 4.5, we applied three unsupervised clustering algorithms (KMC, FCM, and HCA). For comparability across patients, the number of clusters was fixed to six, as this solution provided the best balance between separating histologically recognisable compartments (mite exoskeleton, mite interior/scybala, stratum corneum, dermis, background/debris) and avoiding over-fragmentation of tissue regions. The resulting segmentation maps visualized chemically distinct domains within the tissue.

As shown in Figure 3, the clustering revealed distinct clusters associated with the scabies mite exoskeleton, consistently present across all patient samples. These clusters corresponded to higher absorbance in the 1025–1150 cm^−1^ range, associated with chitin and polysaccharides, and exhibited clear boundaries from adjacent host tissues. We used the concordant detection of mite-associated clusters by at least two algorithms and their co-localisation with mite structures in H&E-stained sections as a qualitative robustness criterion. Importantly, these mite-specific clusters remained detectable even in sections with degraded or inconspicuous structures, highlighting MIA’s sensitivity to low-abundance parasitic elements.

### 2.3. Consistent and Quantitatively Validated Scabies-Associated Absorbance at 1072 cm^−1^ Across Patients

We evaluated the analytical robustness and reproducibility of the 1072 cm^−1^ FTIR spectral marker for *Sarcoptes scabiei* by analyzing FFPE skin tissue from six patients with histologically confirmed scabies. The 1072 cm^−1^ wavenumber corresponds to C–O and C–C stretching vibrations of chitin, a polysaccharide in arthropod exoskeletons but absent in human skin. As shown in Figure 4, FTIR chemical maps reveal intense absorbance at this wavenumber in areas occupied by the mite, appearing as red to orange zones, while the stratum corneum and dermis show lower absorbance due to the absence of chitin. This pattern was consistently observed across all patients, even in samples with sparse mite fragments, demonstrating the robustness of this diagnostic marker.

Figure 5 displays the normalized absorbance values at 1072 cm^−1^, a spectral region associated with chitin-associated C–O/C–C stretching, across the scabies exoskeleton, stratum corneum, and dermis. The scabies exoskeleton showed the highest mean absorbance (0.0207 ± 0.0036), followed by the stratum corneum (0.0148 ± 0.0022) and the dermis (0.0132 ± 0.0021). A two-way ANOVA confirmed a significant main effect of tissue type (F(2, 120) = 220.1, *p* < 0.0001), while inter-sample variability (row factor) was not significant (*p* = 0.9935). Šídák’s post hoc analysis revealed significant differences between all groups, indicating that absorbance at 1072 cm^−1^ reliably distinguishes chitin-rich mite structures from surrounding host tissues. These findings support the use of this spectral feature as a quantitative and reproducible spectral marker for scabies infestation in FFPE sections. 

## 3. Discussion

This preliminary proof-of-concept study demonstrates the technical feasibility and potential diagnostic utility of FTIR microscopy for detecting scabies mites in FFPE human skin sections. Our findings indicate that the spectral features of the scabies exoskeleton—particularly the strong and specific absorbance at 1072 cm^−1^ attributed to chitin-associated C–O/C–C vibrations—provide a scabies-associated molecular marker for intra-lesional detection. These results support the use of FTIR as a label-free, non-destructive analytical adjunct in dermatopathology, but they do not establish patient-level diagnostic performance or superiority over existing methods. Importantly, this specificity currently refers to intra-lesional discrimination between mite exoskeleton and surrounding host tissue rather than to disease-level specificity across different dermatoses, as no healthy skin or non-scabies controls were included. It is important to emphasise that, in our implementation, FTIR microscopy is performed on FFPE sections obtained from routine skin biopsies; as such, it does not reduce the invasiveness of sampling for patients. In this manuscript, we therefore reserve the term “non-invasive” for clinical imaging methods applied directly to the patient (e.g., dermoscopy), and use “non-destructive” to describe the fact that FTIR analysis preserves the integrity of already excised tissue sections.

Conventional diagnostic methods for scabies, such as skin scraping, dermoscopy, and histopathology, are limited by their sensitivity, operator dependence, and the integrity of the samples [6,37,38,39]. Histological confirmation of mites is often rare due to their small size and sparse distribution, and even when present, identification may be confounded by inflammation and tissue artefacts [40]. FTIR microscopy overcomes these issues by probing molecular composition rather than morphology, offering a complementary dimension to structural analysis [13,41].

Within the scope of the present work, FTIR microscopy was not evaluated as an independent diagnostic test in a prospective clinical workflow. Instead, all analysed sections were derived from lesions that had already been diagnosed as scabies by routine histopathology. Consequently, our study was not designed to provide patient-level estimates of diagnostic sensitivity or specificity, nor to allow a head-to-head comparison with dermoscopy, skin scraping, or histology. At this stage, FTIR should therefore be viewed as an adjunctive technique that adds molecular contrast to conventional histopathological assessment in lesions with a high clinical or histological suspicion of scabies, rather than as a validated replacement for existing diagnostic methods.

The spectral fingerprint of the scabies exoskeleton observed in our study aligns with prior research on arthropod cuticles, which consistently show strong absorbance in the 1000–1200 cm^−1^ range due to chitin, a key structural polysaccharide in ectoparasites [42]. Because chitin is highly conserved across arthropods, other mites and ectoparasitic insects are expected to exhibit qualitatively similar carbohydrate-associated bands in this region. Our dataset is restricted to Sarcoptes scabiei, and we did not include other mite species or ectoparasites; therefore, we cannot assess whether the 1072 cm^−1^ signal is specific to scabies mites or constitutes a more general marker of chitin-rich parasitic structures. In the present context, we therefore interpret 1072 cm^−1^ as a scabies-associated band within histologically confirmed scabies lesions, not as a species-specific signature. We fully acknowledge that this carbohydrate-rich region is not chemically unique to chitin: keratinous structures of the stratum corneum, bacterial biofilms, and cellular or keratinous debris may also exhibit prominent C–O and C–O–C stretching vibrations in this spectral window. In our dataset, however, the 1072 cm^−1^ band was significantly more intense and consistently present in regions of interest corresponding to mite exoskeletons than in adjacent stratum corneum and dermis, as summarized in Table 2. Moreover, multivariate image analysis revealed that pixels with high 1072 cm^−1^ absorbance formed clusters that co-localized with morphologically identifiable mite structures rather than with host tissue compartments or non-specific debris. We therefore interpret the 1072 cm^−1^ signal as a chitin-enriched, scabies-associated band in this histological context, while explicitly recognising that it is not an absolutely unique marker of chitin and must be interpreted together with the full spectral pattern and the underlying tissue morphology. This spectral window was absent or minimal in host tissue layers, such as the stratum corneum and dermis, resulting in a distinct diagnostic signature. While dermal collagen and epidermal lipids yielded characteristic Amide I/II and CH_2_ stretches, respectively, they lacked the pronounced carbohydrate signals that define the mite’s molecular architecture. These distinctions are critical in FFPE specimens, where conventional stain uptake or mite morphology may be suboptimal.

Importantly, our MIA approach provided further validation. HCA and KMC clustering robustly classified spectral domains corresponding to mite structures, keratinised epithelium, and collagenous stroma, while FCM captured biochemical transitions at the host–parasite interface. This combination of spectral resolution and spatial mapping enhances diagnostic confidence and opens avenues for automated or semi-automated mite detection algorithms in digital pathology workflows [43,44]. In line with the exploratory and descriptive nature of our MIA, we did not perform cross-validation or formal statistical comparisons between clustering algorithms; instead, we focused on reproducible, visually interpretable segmentations that co-localised with mite structures across multiple methods.

Our results are consistent across six patient samples, highlighting the reproducibility and robustness of the chitin signature at 1072 cm^−1^. Despite inherent variability in tissue composition, fixation, and sectioning, the spectral pattern of the parasite remained conserved—an essential feature for any diagnostic marker. Moreover, the successful application of FTIR to archived FFPE samples underscores its compatibility with routine histopathological workflows and its potential for retrospective analyses or biomarker validation.

A potential differential diagnosis includes *Demodex folliculorum*, a mite commonly found on facial skin. However, scabies typically spares the face and prefers areas such as the interdigital spaces, wrists, and genital region. The absence of facial involvement, therefore, supports the exclusion of scabies. Moreover, although chitin is also a constituent of *Demodex* exoskeletons, its spectral profile and spatial distribution within follicles differ from that of scabies mites. In the present study, we did not include *Demodex* or other ectoparasitic infestations as comparators, and we therefore cannot formally quantify how much the 1072 cm^−1^ marker separates scabies from other chitin-containing parasites at the disease level. The specificity of the 1072 cm^−1^ FTIR peak for *Sarcoptes scabiei* in our study is thus limited to the included, histologically proven scabies lesions and supports its use as a discriminative feature; however, further comparative validation with *Demodex* remains warranted. In addition, scabies can be reliably diagnosed even in the absence of intact tunnels or visible mites. As highlighted by Zelger, parakeratotic stratum corneum with spike imprint—left behind by mite spines—constitutes a diagnostic clue even when other parasitic elements are absent [45].

Nonetheless, certain limitations should be acknowledged. First, this is a small proof-of-concept study based on six patients, all of whom had histologically confirmed scabies, and FTIR microscopy was only applied to sections that were already positive on routine histology. We did not include healthy skin, inflammatory dermatoses without scabies, or other parasitic infestations as negative or disease controls; instead, we used stratum corneum and dermis from the same sections as intra-sample reference tissues. These reference compartments demonstrate that FTIR can differentiate chitin-rich mite structures from surrounding skin, but do not constitute true negative controls and therefore cannot provide information on diagnostic specificity at the disease level. Also, we did not perform a formal, head-to-head comparison with dermoscopy, skin-scraping microscopy, or histopathology. In addition, we did not include other arthropod parasites (e.g., Demodex mites, lice, fleas, or ticks) as comparators. As chitin is a common structural component of arthropod exoskeletons, FTIR bands in the 1000–1200 cm^−1^ region are unlikely to be unique to *Sarcoptes scabiei*. Our data therefore do not allow us to determine whether the observed spectral pattern differentiates scabies mites from other chitin-rich ectoparasites. The current marker should be regarded as indicating the presence of chitin-rich parasitic structures in the appropriate clinical and histopathological context, rather than as a species-specific fingerprint. Therefore, no formal conclusions regarding clinical diagnostic reliability (e.g., sensitivity, specificity, predictive values or overall accuracy) can be drawn from this dataset, and our data do not permit calculation of diagnostic sensitivity or specificity for FTIR at the patient level, nor do they allow quantitative comparison with the performance of current routine methods or systematic evaluation of FTIR in cases that are false-negative by dermoscopy or histology. Consequently, no formal conclusions regarding diagnostic reliability (e.g., sensitivity, specificity, predictive values, or clinical accuracy) can be drawn from this small cohort, and our findings should be regarded as hypothesis-generating.

Our conclusions regarding “specificity” are restricted to the intra-lesional discrimination between chitin-rich mite structures and surrounding human skin compartments within known scabies lesions. Second, FTIR microscopy, although non-destructive and label-free, is limited by its spatial resolution (~2.6 µm) and spectral overlap in highly heterogeneous tissues. Inflammatory infiltrates or necrotic debris may confound spectral interpretation, though multivariate clustering helps mitigate these effects. Despite the presence of an inflammatory infiltrate, we were able to detect chitin in all samples, indicating good analytical sensitivity for chitin-rich structures within histologically positive lesions. Third, we did not systematically annotate or quantify bacterial biofilms, dense keratinous crusts, or other carbohydrate-rich deposits within our regions of interest. Such structures could, in principle, contribute to absorbance in the 1000–1200 cm^−1^ range and generate false-positive signals if they were mis-registered as parasite-associated. Careful correlation with the corresponding histological sections and the use of multivariate image analysis were therefore essential to minimise this risk in the present study, but more targeted controls will be required to characterise these potential confounders in detail. Future studies should explicitly include healthy and disease controls, prospectively collect samples from suspected scabies cases, and incorporate parallel dermoscopy, skin scraping, and histology in order to calculate diagnostic sensitivity and specificity and to quantify whether FTIR can rescue cases that are false-negative by routine methods.

## 4. Materials and Methods

### 4.1. Reagents and Materials

Xylene, ethanol, hematoxylin (Mayer’s solution), and eosin (Eosin 0.5% aqueous) were obtained from MORPHISTO GmbH (Offenbach, Germany). Calcium fluoride (CaF_2_) slides (1.0 mm thickness; KORTH KRISTALLE GmbH, Altenholz, Germany) and adhesive glass slides (1.0 mm thickness; Klinipath VWR International B.V., Amsterdam, The Netherlands) were used in the study.

### 4.2. Acquisition and Preparation of Skin Tissue Specimens

Formalin-fixed, paraffin-embedded (FFPE) skin tissue samples were obtained from the biobank of the Department of Dermatology, Innsbruck. All specimens were collected with written informed consent and ethical approval from the Ethics Committee of the Medical University of Innsbruck (EK 1011/2025). Diagnosis of scabies was established through routine hematoxylin and eosin (H&E) staining and the histological identification of parasitic elements.

### 4.3. FTIR Microscopy for Tissue Characterisation

Two adjacent tissue sections, each 5 µm thick, were prepared from each selected paraffin block. One was stained with H&E for histological evaluation, while the other remained unstained for spectroscopic imaging on CaF_2_ slides (KORTH KRISTALLE GmbH, Altenholz, Germany). Unstained sections were deparaffinized using xylol for 45 min, followed by 100% ethanol for 5 min, then 70% ethanol for 5 min, and finally 50% ethanol for 5 min. They were then dried in an incubator (60 °C) for 15 min and at room temperature (20–22 °C) for 30 min. Fourier-transform infrared (FTIR) spectroscopic imaging was performed at room temperature in transmission mode using a Bruker Vertex 70 FTIR spectrometer (Bruker Optik GmbH, Ettlingen, Germany) coupled to a Hyperion 3000 microscope (Bruker Optik GmbH, Ettlingen, Germany), equipped with a focal plane array (FPA) detector comprising 64 × 64 MCT-D364 elements.

The system was purged with dry air to reduce atmospheric interference. FTIR spectral data were collected with a resolution of 2.65 µm and 16 co-added scans across a range of 3900 to 850 cm^−1^. This descending order reflects standard FTIR convention, where wavenumbers decrease from left to right, corresponding to a shift from higher to lower energy vibrations. This orientation facilitates interpretation and aligns with established practices in infrared spectral analysis. Background spectra were recorded before each measurement.

### 4.4. Spectral Data Processing and Image Assembly

Spectral data processing and image assembly were performed using OPUS 6.5 (Bruker) and the CytoSpec software package (www.cytospec.com, accessed on 27 October 2025, Hamburg, Germany). Univariate chemical maps were generated using Quasar software (Version 1.7.0) [46].

### 4.5. FTIR Imaging Analysis

FTIR microscopic images were imported into Quasar for univariate analysis. Spectra were area-normalized, baseline-corrected, and Gaussian-smoothed (standard deviation: 2) across the spectral range of 3700–850 cm^−1^. These preprocessing steps enhanced spectral resolution, removed background slopes, and minimized variations related to tissue density and surface irregularities [12]. Chemical maps were generated based on the integrated absorbance of 30 selected bands, as listed in [25]. This approach enabled a computationally efficient yet representative visualization of key spectral features [33,34].

Multivariate image analysis (MIA) of FTIR data was conducted using CytoSpec™ software (version 2.00.08) within the 1000–1200 cm^−1^ spectral range, highlighting the vibrational bands of carbohydrate-rich structures, such as chitin. Three unsupervised clustering algorithms classified tissue compartments by molecular composition. For MIA, spectra were cropped to 1000–1200 cm^−1^, and the same preprocessing steps (area normalization, baseline correction, Gaussian smoothing with a standard deviation of 2) were applied prior to clustering. We used three unsupervised clustering algorithms—k-means clustering (KMC), fuzzy c-means clustering (FCM), and hierarchical cluster analysis (HCA)—all with an Euclidean distance metric. For KMC and FCM, the number of clusters was fixed to six for all datasets. This value was chosen after exploratory runs with 3–10 clusters and selecting the solution that best matched the histologically recognisable compartments (mite exoskeleton, mite interior/scybala, stratum corneum, dermis, and background/debris) while avoiding over-segmentation. For HCA, the dendrogram was cut at six clusters to enable direct comparison with KMC/FCM segmentations. Because our aim was descriptive tissue segmentation rather than training a predictive classifier, no cross-validation in the machine learning sense was performed. Likewise, we did not conduct formal statistical tests to compare clustering algorithms; instead, robustness was assessed qualitatively by requiring that mite-associated domains were reproduced by at least two algorithms and co-localised with mite structures on the corresponding light micrographs and H&E-stained sections. Clustering results were validated using light micrographs and H&E-stained sections, which segmented the tissue into biochemically distinct regions associated with the scabies mite.

### 4.6. Statistical Analysis of Absorbance Values

Statistical analyses were performed using GraphPad Prism version 10.0.2 (GraphPad Software, San Diego, CA, USA). Absorbance values at 1072 cm^−1^ were assessed for normality using the Shapiro–Wilk test. Group differences were evaluated using a two-way ANOVA with tissue type as the main factor. Post hoc comparisons were conducted using Šídák’s multiple comparisons test. A two-tailed significance level of 0.05 and a 95% confidence interval were applied throughout. Adjusted *p*-values < 0.05 were considered statistically significant.

## 5. Conclusions

In this proof-of-concept study, we show that FTIR microscopy can reliably identify *Sarcoptes scabiei* structures in FFPE human skin sections based on their molecular composition rather than morphology. Given the intentionally small, homogeneous cohort (n = 6), these results should be interpreted as preliminary evidence of technical feasibility and analytical robustness, rather than as establishing clinical diagnostic reliability. In particular, the chitin-associated absorbance at 1072 cm^−1^ emerged as a robust, quantitatively validated spectral marker that consistently distinguished scabies exoskeletons from the surrounding stratum corneum and dermis across all examined patients. Multivariate image analysis of the 1000–1200 cm^−1^ fingerprint region further enabled precise spatial mapping of mite structures and host–parasite interfaces, even when parasitic elements were small, fragmented, or inconspicuous on conventional staining.

Taken together, these findings position FTIR microscopy as a label-free, non-destructive adjunct to routine histopathology for the diagnosis of scabies in tissue sections. By adding molecular contrast centered on chitin-associated signatures, FTIR has the potential to increase diagnostic confidence in challenging or ambiguous cases and to support retrospective studies using archived FFPE material.

Future work should extend this approach to larger, prospectively collected cohorts, incorporate healthy skin and a spectrum of non-scabies inflammatory and parasitic dermatoses as negative and disease controls, include comparative analyses with other ectoparasites, and explore the integration of automated spectral classification into digital pathology workflows. Such developments will be essential to rigorously assess diagnostic sensitivity and specificity at the patient level and could help to translate FTIR microscopy from an experimental technique into a practical tool that complements existing clinical and histopathological methods for parasitic skin diseases. Until such studies are available, our findings should be interpreted as analytical proof-of-concept for intra-lesional mite detection in FFPE sections rather than as evidence for a superior clinical diagnostic performance compared with dermoscopy or routine histology.

## Figures and Tables

**Figure 1 ijms-26-11597-f001:**
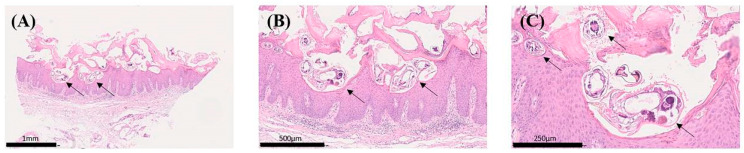
Histological features of scabies norvegica in H&E-stained tissue sections: (**A**) Overview showing hyperkeratosis with embedded mite structures (arrows). (**B**) Intermediate magnification revealing multiple mites in the stratum corneum and underlying inflammatory cell infiltrate. (**C**) High magnification detailing the cuticular architecture of scabies. Scale bars are shown in the images. Image taken from one of the case studies included in this report. This case was selected as a reference for spectral calibration due to the ideal preservation and abundance of mites—an occurrence that is exceptionally rare in routine histopathology. H&E, hematoxylin and eosin.

**Figure 2 ijms-26-11597-f002:**
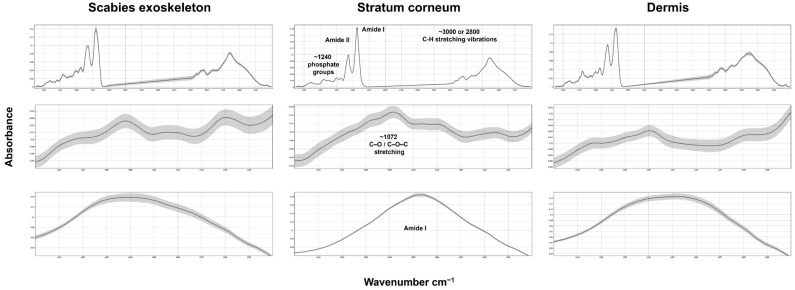
Representative FTIR absorbance spectra from histologically confirmed scabies lesions showing characteristic spectral differences between tissue compartments. Each panel illustrates mean spectra (solid line) ± standard deviation (shaded area) for regions of interest corresponding to mite exoskeleton, mite interior/scybala, stratum corneum, and dermis. Prominent vibrational bands are indicated: the Amide I band (~1650 cm^−1^) associated with protein C=O stretching; Amide II (~1550 cm^−1^) from N–H bending/C–N stretching; and the 1000–1200 cm^−1^ carbohydrate region encompassing the chitin-associated C–O/C–C stretching vibration (~1072 cm^−1^). Insets highlight the Amide I and carbohydrate regions to visualise amplitude and line-shape differences between parasite and host tissue.

**Figure 3 ijms-26-11597-f003:**
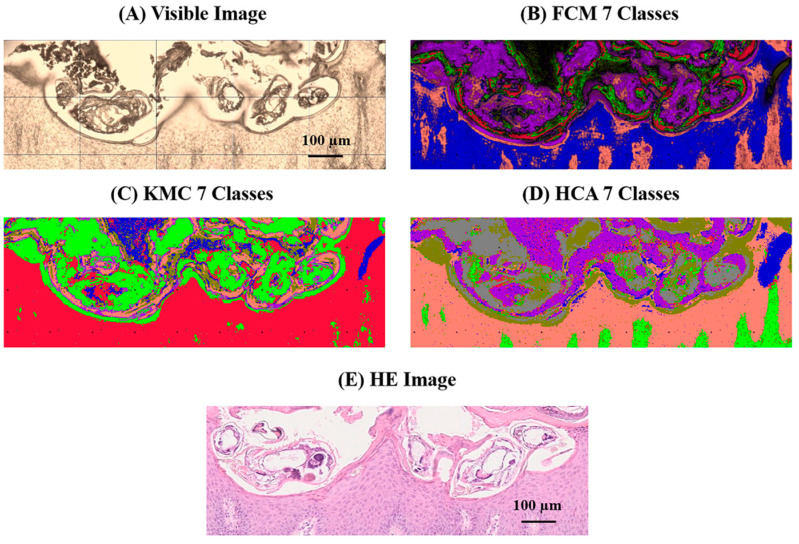
Multivariate image analysis (MIA) of scabies-infected skin tissue using FTIR imaging: (**A**) Visible light image of an FFPE skin section with *Sarcoptes scabiei* infestation. (**B**–**D**) Fuzzy c-means clustering (FCM), k-means clustering (KMC), and hierarchical cluster analysis (HCA) maps from the 1000–1200 cm^−1^ spectral interval. Clusters corresponding to the scabies mite’s body wall, legs, and internal structures are spatially separated from those representing the stratum corneum, dermis, and background, making the different tissue compartments visually distinct. (**E**) Hematoxylin and eosin (H&E)-stained section of the same region confirming the anatomical localisation of mite structures. Together, these panels illustrate how MIA delineates parasite-associated domains on the basis of carbohydrate-rich vibrational signatures in the 1000–1200 cm^−1^ range (scale bar: 100 µm). Abbreviations: MIA, multivariate image analysis; FTIR, Fourier-transform infrared; FFPE, formalin-fixed paraffin-embedded; FCM, fuzzy c-means; KMC, k-means clustering; HCA, hierarchical cluster analysis; H&E, hematoxylin and eosin.

**Figure 4 ijms-26-11597-f004:**
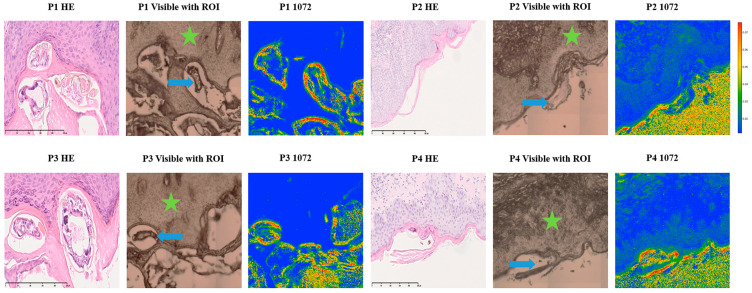
FTIR microscopy shows consistent chitin-associated absorbance in scabies mite structures across four representative human skin samples. For each sample, the left panel depicts the corresponding hematoxylin and eosin (H&E)-stained section, the middle panel shows the visible light micrograph of the formalin-fixed paraffin-embedded (FFPE) skin section, and the right panel displays the univariate FTIR chemical map at 1072 cm^−1^. Arrows indicate the scabies exoskeleton, whereas stars mark dermal reference areas used for comparison. High absorbance at 1072 cm^−1^ in mite structures and lower absorbance in dermis and stratum corneum illustrate the preferential enrichment of this band in chitin-rich parasite regions, supporting its diagnostic utility as a scabies-associated spectral marker rather than a chemically unique signal (scale bar: 200 µm). Abbreviations: FTIR, Fourier-transform infrared; FFPE, formalin-fixed paraffin-embedded; H&E, hematoxylin and eosin.

**Figure 5 ijms-26-11597-f005:**
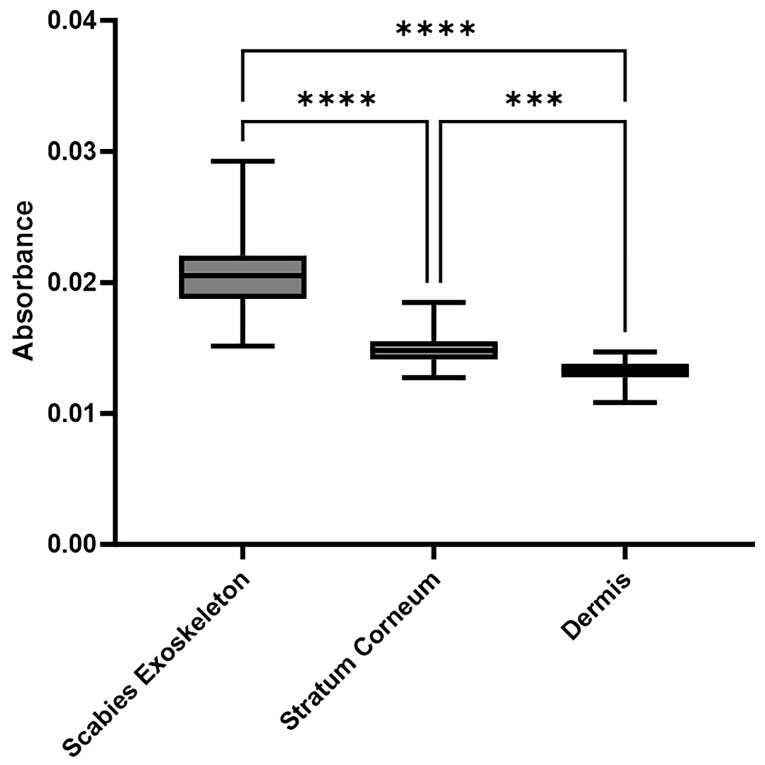
FTIR analysis of scabies-infected skin tissue. Box-and-whisker plots summarize absorbance at 1072 cm^−1^ for regions of interest placed in the mite exoskeleton, stratum corneum, and dermis across all included samples, indicating strong spectral contrast between chitin-rich parasite structures and the surrounding host tissue. Mite regions showed significantly higher 1072 cm^−1^ absorbance compared to adjacent reference tissue compartments (stratum corneum and dermis) from the same sections, illustrating the spectral contrast between chitin-rich parasite structures and surrounding host skin. Data are shown as mean ± standard deviation (SD) over all measurements per tissue type; a two-way analysis of variance (ANOVA) with Šídák’s post hoc test was used to assess differences between groups. Asterisks denote levels of statistical significance (*** *p* < 0.001, **** *p* < 0.0001). Abbreviations: FTIR, Fourier-transform infrared; SD, standard deviation; ANOVA, analysis of variance.

**Table 1 ijms-26-11597-t001:** Clinical–pathological characteristics of the group by proven infection detected histologically.

NO	Biopsie Site	Sex	Age	Histology
1	left thigh	M	73	positive
2	gluteal	M	81	positive
3	lumbal	M	36	positive
4	not specified	F	68	positive
5	not specified	F	88	positive
6	abdomen	F	21	positive

NO: case number; Sex: female (F), male (M); positive = proven infection detected histologically.

**Table 2 ijms-26-11597-t002:** Key FTIR spectral marker bands across the tissue regions, emphasising the distinctive chitin-associated peaks in the scabies exoskeleton. Relative signal intensities are indicated by increasing numbers of checkmarks (✓). ✓ = present; ✓✓ = strong; ✓✓✓ = very strong; — = not prominent.

Wavenumber (cm^−1^)	Vibrational Mode	Dermis	Stratum Corneum	Scabies Exoskeleton	Interpretation
~3300	OH/NH stretch	✓✓✓	✓✓	✓✓✓	protein amide A vibration [27,28]
~2920	CH_2_ asym. stretch	✓	✓✓✓	✓✓	Lipid chains (membranes, cuticle) [29]
~2850	CH_2_ sym. stretch	✓	✓✓✓	✓✓	Lipid methylene groups [29]
~1730	C=O stretch (ester)	—	✓✓	✓	Triacylglycerol:Tripetroselinin (TPE) [30]
~1650	Amide I (C=O)	✓✓✓	✓	✓✓✓	Protein secondary structure [31,32]
~1550	Amide II (N-H/C-N)	✓✓✓	✓	✓✓✓	Proteins with α-helix and random coil conformations [33,34]
~1200–1000	C–O/C–C stretch	—	✓	✓✓✓	Chitin-rich carbohydrate (arthropod exoskeleton component) [35,36]

## Data Availability

The original contributions presented in this study are included in the article. Further inquiries can be directed to the corresponding authors.

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
