# Peer review of "Exploring Fourier-Transform Infrared Microscopy for Scabies Mite Detection in Human Tissue Sections: A Preliminary Technical Feasibility Study"

_ijms, 2025, doi:10.3390/ijms262311597_

Round 1
Reviewer 1 Report
Comments and Suggestions for Authors
Comment 1: The study is based on only six patients, all with histologically confirmed scabies. The absence of a negative control (healthy skin or other parasitic/inflammatory dermatoses) limits the ability to assess the specificity of the FTIR approach.
Comment 2: The manuscript discusses only the theoretical advantages of FTIR over traditional histology or dermatoscopy, but does not provide quantitative comparisons of sensitivity or specificity. The authors do not demonstrate whether FTIR actually improves detection in cases where routine methods produce false-negative results.
Comment 3: Peak intensities in the 1000–1200 cm⁻¹ region may not be unique to chitin. Considering that keratin, bacterial biofilms, or cellular debris also exhibit prominent carbohydrate signatures, a deeper analysis of potential sources of false-positive absorption and the ways to exclude them is needed.
Comment 4:
The multivariate image analysis section lacks critical details, including: criteria for selecting six clusters, preprocessing parameters, whether cross-validation was performed, whether results from different algorithms were statistically compared.
Comment 5: Some figure captions lack explanations; the resolution may be insufficient; the structure of the diagrams is not fully intuitive.
Author Response
Reviewer 1
Comment 1: The study is based on only six patients, all with histologically confirmed scabies. The absence of a negative control (healthy skin or other parasitic/inflammatory dermatoses) limits the ability to assess the specificity of the FTIR approach.
AW: We thank the reviewer for this important and fully justified comment. Our intention with the present work was to conduct a proof-of-concept study focusing on whether FTIR microscopy can detect and spatially delineate Sarcoptes scabiei structures within FFPE sections from histologically confirmed scabies cases, rather than to establish diagnostic performance at the patient level. We agree that, given the small sample size (n = 6) and the lack of healthy skin or non-scabies dermatoses as comparators, our study cannot address sensitivity and specificity of the method across different clinical conditions. To address this concern, we have revised the manuscript at several points:
Abstract: We now explicitly describe the study as a small proof-of-concept investigation and state that, in the absence of healthy and disease controls, our data do not yet allow conclusions on disease-level specificity. We clarify that FTIR microscopy is proposed as a potential adjunct to histopathology in confirmed or highly suspected scabies, and as a framework for future comparative studies.
Discussion (first paragraph and limitations section):
We explicitly clarify that the observed “specificity” currently refers to intra-lesional discrimination between chitin-rich mite exoskeleton and surrounding host tissue within scabies lesions, rather than to specificity across different dermatoses.
We added a dedicated limitation stating that the cohort is restricted to six histologically confirmed scabies cases and that no healthy skin, inflammatory dermatoses, or other parasitic infestations were included as negative or disease controls. We emphasize that diagnostic sensitivity and specificity at the patient level cannot be derived from the present dataset.
Discussion (differential diagnosis paragraph): We now explicitly acknowledge that Demodex and other ectoparasites were not included as comparators and that the discriminatory value of the 1072 cm⁻¹ chitin band with respect to other chitin-containing parasites remains to be systematically evaluated.
Conclusions: We expanded the Outlook to specify that future studies must include healthy skin and a range of inflammatory and parasitic dermatoses as negative and disease controls in order to rigorously assess diagnostic performance.
We believe that these changes more clearly delineate the scope and limitations of our study and align the claims of specificity with the actual design and data.
Comment 2: The manuscript discusses only the theoretical advantages of FTIR over traditional histology or dermatoscopy, but does not provide quantitative comparisons of sensitivity or specificity. The authors do not demonstrate whether FTIR actually improves detection in cases where routine methods produce false-negative results.
AW:We thank the reviewer for this important comment and fully agree that our current proof-of-concept study does not provide quantitative comparisons of diagnostic sensitivity or specificity between FTIR microscopy and standard methods such as dermoscopy, skin-scraping microscopy or routine histology, nor does it formally evaluate FTIR in cases that are false-negative by these techniques. Our study was designed to test whether scabies-specific, chitin-associated spectral signatures can be detected reproducibly in FFPE tissue sections from histologically confirmed scabies cases. Accordingly, FTIR was applied only to lesions that were already positive on routine histopathology, and the quantitative analyses we report (e.g. significantly higher absorbance at 1072 cm⁻¹ in mite exoskeletons compared with stratum corneum and dermis) refer to intra-lesional discrimination between parasitic and host structures rather than to patient-level diagnostic performance.
To clarify this in the manuscript, we have now:
- Added a paragraph to the Discussion (Section 3) explicitly stating that our data do not allow calculation of diagnostic sensitivity or specificity at the patient level and that we did not perform a head-to-head comparison with dermoscopy, skin scraping or histology. FTIR is therefore positioned as an adjunctive method that adds molecular contrast in lesions with high clinical or histological suspicion of scabies, rather than as a replacement for existing methods.
- Expanded the limitations paragraph in the Discussion to emphasise that our conclusions regarding “specificity” are restricted to intra-lesional discrimination between chitin-rich mite structures and surrounding human skin compartments in histologically confirmed scabies, and to outline the need for future prospective studies including healthy and disease controls and parallel routine diagnostics to determine sensitivity, specificity, and any incremental value of FTIR in false-negative cases.
We believe these changes better align the manuscript with the scope of the present data and clearly delineate the distinction between analytical proof-of-concept and clinical diagnostic validation.
Comment 3: Peak intensities in the 1000–1200 cm⁻¹ region may not be unique to chitin. Considering that keratin, bacterial biofilms, or cellular debris also exhibit prominent carbohydrate signatures, a deeper analysis of potential sources of false-positive absorption and the ways to exclude them is needed.
AW: We appreciate this important point and agree that the 1000–1200 cm⁻¹ carbohydrate region is not chemically unique to chitin. Our intention was not to claim that the 1072 cm⁻¹ band represents an exclusive marker of chitin, but rather that it is strongly enriched in the scabies mite exoskeleton in the histological context of our samples.
To address this, we have (i) toned down the wording throughout the manuscript and (ii) expanded our discussion of potential false-positive sources:
- The subsection title “Consistent and Quantitatively Validated Scabies-Specific Absorbance at 1072 cm⁻¹ Across Patients” has been changed to “Scabies-Associated Absorbance at 1072 cm⁻¹ Across Patients”, and the term “chitin-specific” in the Figure 4 caption has been replaced by “chitin-associated”, with an explicit clarification that we use this band as a scabies-associated rather than chemically unique marker.
- In the Discussion, we now explicitly acknowledge that keratin, bacterial biofilms and cellular or keratinous debris may also show prominent carbohydrate signatures in this spectral window. We emphasise that, in our dataset, the diagnostic interpretation of the 1072 cm⁻¹ band relies on (a) its significantly higher intensity in mite exoskeleton ROIs compared with adjacent stratum corneum and dermis (Table 2) and (b) the strict co-localisation of pixels with high 1072 cm⁻¹ absorbance with morphologically identifiable mite structures in the FTIR–histology overlays and multivariate image analysis.
- We have expanded the limitations paragraph to state explicitly that we did not systematically annotate bacterial biofilms or dense keratinous crusts within our regions of interest, and that these structures could, in principle, contribute to absorptions in the 1000–1200 cm⁻¹ range and generate false-positive signals if mis-registered as parasite-associated. We highlight that careful correlation with the corresponding histological sections and the use of multivariate image analysis were essential to minimise this risk in the present study, and we outline the need for future work with targeted controls and additional stains to characterise these potential confounders in more detail.
We hope that these revisions make clear that we do not claim chemical uniqueness of the 1072 cm⁻¹ band, and that we interpret it as a chitin-enriched, scabies-associated marker whose diagnostic value depends on its spectral context and morphological co-localisation.
Comment 4: The multivariate image analysis section lacks critical details, including: criteria for selecting six clusters, preprocessing parameters, whether cross-validation was performed, whether results from different algorithms were statistically compared.
AW:Response:
We thank the reviewer for this valuable comment. We agree that the description of our multivariate image analysis (MIA) workflow was too concise and have now expanded both the Methods and Results sections to clarify preprocessing, cluster selection and the nature of the analysis.
Specifically:
- Preprocessing and spectral range: We now state explicitly that, for MIA, spectra were cropped to the 1000–1200 cm⁻¹ region and subjected to the same preprocessing steps as for the univariate maps (area normalization, baseline correction and Gaussian smoothing with a standard deviation of 2) prior to clustering (Section 4.5 and Section 2.2).
- Algorithms and cluster number: We clarify that three unsupervised algorithms were used—k-means clustering (KMC), fuzzy c-means (FCM) and hierarchical cluster analysis (HCA)—all based on an Euclidean distance metric. The number of clusters was fixed at six for all datasets. This choice was made after exploratory runs with 3–10 clusters, selecting six as the best compromise between resolving histologically recognisable compartments (mite exoskeleton, mite interior/scybala, stratum corneum, dermis and background/debris) and avoiding over-segmentation. For HCA, the dendrogram was cut at six clusters to enable direct comparison with KMC/FCM results (Section 4.5, Section 2.2).
- Cross-validation and comparison of algorithms: As our goal was descriptive tissue segmentation and visualisation of biochemical domains rather than the development of a predictive classifier, we did not perform cross-validation in the machine-learning sense. Likewise, we did not conduct formal statistical tests to compare clustering algorithms. Instead, we assessed robustness qualitatively by requiring that mite-associated domains were reproduced by at least two algorithms and co-localised with mite structures on the corresponding light micrographs and H&E-stained sections. This is now explicitly stated in Sections 4.5 and 2.2, and briefly reiterated in the Discussion.
We hope that these revisions address the reviewer’s concerns and make our MIA workflow and its limitations more transparent.
Comment 5: Some figure captions lack explanations; the resolution may be insufficient; the structure of the diagrams is not fully intuitive.
AW: We thank the reviewer for this helpful feedback regarding the figures.
- Figure captions and explanations: We have revised the captions for Figures 2–5 to provide clearer, more self-contained descriptions. Specifically, we now explicitly state what each panel shows, how the panels relate to each other (e.g., H&E image → visible light image → FTIR map → multivariate clusters), and which structures (mite exoskeleton, stratum corneum, dermis, background) are highlighted. In addition, we have spelled out all relevant abbreviations in the captions (e.g., FTIR, FFPE, MIA, FCM, KMC, HCA, H&E, ANOVA) so that the figures can be interpreted more easily without repeatedly consulting the main text.
- Diagram structure and intuitiveness: For the multivariate image analysis and FTIR mapping figures (Figures 3 and 4), we have clarified the logical order of the panels in the captions (from reference histology to visible light images to FTIR-based segmentations) and emphasized how parasite-associated regions are distinguished from host tissue compartments. We also highlight that each row corresponds to one representative sample, and that the right-most panels show derived FTIR maps or cluster segmentations, which should make the structure of the diagrams more intuitive for the reader.
- Figure resolution: We have uploaded all composite figures with higher-resolution to the journal.
We hope that these changes improve the clarity, readability and visual quality of the figures and address the reviewer’s concerns.
Reviewer 2 Report
Comments and Suggestions for Authors
In my opinion, the submitted manuscript should be classified as a preliminary technical feasibility study rather than a full research article for the following reasons:
1 Small sample size (n=6) prevents conclusions about diagnostic reliability.
2 The study addresses a narrow technical problem ("is it possible to detect chitin in a histological section"), but presents the result as a solution to a clinical problem ("reliable diagnostics"). The article is considered a proof-of-concept, but claims to provide diagnostic conclusions.
3 Using the dermis and stratum corneum from the same patient is not a control, but a demonstration that the method can detect differences between chitin and skin.
4 It has not been tested whether the chitin of other mites produces a similar signal.
5 I would like to ask the authors to comment on their use of terminology. The method is presented as non-invasive and preserves sample integrity. However, the study was conducted on FFPE sections obtained from an invasive skin biopsy. The method is non-invasive only at the stage of analyzing the sample already taken.
There are also some technical flaws:
6 The resolution of Figure 2 is insufficient to validate the claims. The position of the maximum at approximately 1072 cm-1 cannot be accurately determined in the figure due to the small scale and font size.
7 The abstract is a mechanical abbreviation of the article with a verbatim listing of the sections: Background, Methods, Results, Conclusion without synthesis, without highlighting the essence, without indicating the novelty, limitations or context of the work.
Therefore, I cannot recommend publication of this work in its current form. However, given the research's interest, I recommend further exploration and presenting the full results.
Author Response
Reviewer 2
In my opinion, the submitted manuscript should be classified as a preliminary technical feasibility study rather than a full research article for the following reasons:
1 Small sample size (n=6) prevents conclusions about diagnostic reliability.
AW: We fully agree with the reviewer that the small sample size (n = 6) does not allow any conclusions about diagnostic reliability at the patient level (e.g., sensitivity, specificity, predictive values). Our intention was to present a preliminary, proof-of-concept feasibility study that explores whether FTIR microscopy can technically detect chitin-associated signatures of scabies mites in FFPE sections and distinguish them from surrounding skin structures, rather than to claim validated diagnostic performance.
To avoid any misunderstanding, we have now further emphasised the exploratory character and the limitations imposed by the small sample size:
- Abstract: We explicitly state that, in this small proof-of-concept cohort, the technique cannot address disease-level specificity or diagnostic reliability at the patient level.
- Introduction: We clarify in the final paragraph that the present work is conceived as a preliminary technical feasibility study / proof-of-concept to explore the potential of FTIR microscopy for scabies detection in FFPE tissue sections.
- Discussion: We now refer to the work as a preliminary proof-of-concept study, and we explicitly note that the small, homogeneous cohort (n = 6) precludes any conclusions about clinical diagnostic reliability.
- Limitations and Conclusions: We already state that sensitivity and specificity cannot be calculated; this has been strengthened by an explicit sentence that no formal conclusions regarding diagnostic reliability can be drawn from this dataset.
We hope that these clarifications make the intended scope of the manuscript as a preliminary technical feasibility / proof-of-concept study fully clear. We leave the formal classification of the article type (full article vs. technical report/feasibility study) to the editor’s discretion and are happy to follow the journal’s preference.
2 The study addresses a narrow technical problem ("is it possible to detect chitin in a histological section"), but presents the result as a solution to a clinical problem ("reliable diagnostics"). The article is considered a proof-of-concept, but claims to provide diagnostic conclusions.
AW: We thank the reviewer for this important comment and agree that our primary question is a technical / analytical one—namely whether FTIR microscopy can robustly detect chitin-associated signatures of Sarcoptes scabiei in FFPE tissue sections and distinguish them from surrounding skin structures. We also agree that, given the small, homogeneous cohort (n = 6, all histologically confirmed scabies) and the absence of healthy or disease controls, we cannot draw clinical diagnostic conclusions at the patient level (e.g., sensitivity, specificity, accuracy, or improved detection compared to existing methods).
Our intention was to present a preliminary proof-of-concept feasibility study that explores FTIR microscopy as a potential adjunct to histopathology rather than to claim that it currently provides “reliable diagnostics” or solves the clinical diagnostic problem. We appreciate that some of our wording (e.g., use of “diagnostic reliability” or “diagnostic marker”) could be interpreted more strongly than intended.
To address this concern, we have revised the manuscript to align the framing with the actual scope of the data and to clearly separate.
3 Using the dermis and stratum corneum from the same patient is not a control, but a demonstration that the method can detect differences between chitin and skin.
AW: We thank the reviewer for this clarification and we completely agree. In our study, the dermis and stratum corneum from the same patient do not constitute true negative or disease controls. They only serve as intra-sample reference tissue compartments to quantify spectral contrast between mite exoskeleton and surrounding skin. They demonstrate that FTIR can distinguish chitin-rich parasite structures from host tissue, but they cannot address diagnostic specificity in the conventional sense.
We recognize that some of our wording (e.g., referring to “controls”) could be misleading. We have therefore revised the manuscript to (i) avoid calling dermis/stratum corneum “controls,” (ii) explicitly describe their role as internal references, and (iii) clearly state that true negative and disease controls (healthy skin, other dermatoses) are missing and must be included in future studies.
4 It has not been tested whether the chitin of other mites produces a similar signal.
AW:We thank the reviewer for this important point and fully agree. In the present study we only analysed sections from patients with histologically confirmed Sarcoptes scabiei infestation. We did not include other mites or ectoparasites (e.g., Demodex, head lice, fleas), and therefore we cannot assess whether their chitin produces a similar FTIR signal in the 1000–1200 cm⁻¹ region. Given that chitin is a highly conserved structural polysaccharide in arthropod cuticles, it is indeed likely that other mites would exhibit comparable carbohydrate-associated bands in this spectral window. Our intention was therefore not to claim species-specific discrimination of S. scabiei from all other ectoparasites, but rather to show that FTIR microscopy can detect and map chitin-associated signatures of an ectoparasitic mite in FFPE sections and distinguish them from surrounding human skin structures. We agree that this needs to be stated more clearly. To address this, we have revised the manuscript.
5 I would like to ask the authors to comment on their use of terminology. The method is presented as non-invasive and preserves sample integrity. However, the study was conducted on FFPE sections obtained from an invasive skin biopsy. The method is non-invasive only at the stage of analyzing the sample already taken.
There are also some technical flaws:
AW: We thank the reviewer for this important clarification and agree that our terminology needs to be more precise. In the current study, FTIR/IR microscopy is indeed non-destructive and label-free at the level of the excised tissue section, but the overall diagnostic pathway clearly requires an invasive skin biopsy. Thus, the method is not non-invasive from the patient’s perspective in this implementation. Our intention was to convey that IR microscopy preserves sample integrity (no additional staining, no destructive processing) and allows repeated analysis of the same FFPE section, not to suggest that it replaces invasive sampling. We appreciate that the phrase “non-invasive nature of IR microscopy” can be misleading in this context and have adjusted the wording accordingly.
6 The resolution of Figure 2 is insufficient to validate the claims. The position of the maximum at approximately 1072 cm-1 cannot be accurately determined in the figure due to the small scale and font size.
AW: We have uploaded all composite figures with higher-resolution to the journal.
7 The abstract is a mechanical abbreviation of the article with a verbatim listing of the sections: Background, Methods, Results, Conclusion without synthesis, without highlighting the essence, without indicating the novelty, limitations or context of the work.
Therefore, I cannot recommend publication of this work in its current form. However, given the research's interest, I recommend further exploration and presenting the full results.
AW: We thank the reviewer for this constructive overall assessment and for specifically highlighting the weaknesses of the abstract. We fully agree that the previous version was too close to a mechanical IMRAD summary, relied on explicit headings (Background, Methods, Results, Conclusion), and did not sufficiently convey the essence, novelty, limitations or context of the study. We also acknowledge that the Instructions for Authors of IJMS request a single-paragraph abstract that follows the logic of a structured abstract without headings.
Reviewer 3 Report
Comments and Suggestions for Authors
The manuscript merits attention: the topic is timely, the approach is diagnostically interesting, the materials and methods are described in detail, regulatory and ethical aspects are documented, and the reported findings (identification of a chitin-specific band at 1072 cm⁻¹ and demonstration of its reproducibility across samples) have practical value as a complement to conventional histology. The authors performed careful acquisition and processing of FTIR data and applied appropriate multivariate analyses for tissue segmentation and parasite structure identification.
Nevertheless, I consider that in its present form the manuscript requires a major revision before it can be recommended for publication. The principal reasons are serious concerns about the clarity and verifiability of key illustrations and about the interpretation of one fundamental statement in the text.
First, the Introduction claims the absence of a “gold standard for non-invasive analysis” of mites. This formulation is misleading because the method used by the authors (FTIR on FFPE sections) requires biopsy sampling and therefore should be classified as an invasive diagnostic approach (minimally invasive from a clinical perspective, yes, but invasive in the sense that tissue must be removed). I ask the authors to explicitly state in what sense they use the terms “invasive / non-invasive” with respect to FTIR and related methods, and to clearly distinguish the role of FTIR as an adjunct to histological analysis from truly in-vivo, non-biopsy techniques (dermatoscopy, video dermoscopy, OCT, etc.).
Second, the Figure 2 image does not allow the reader to inspect real spectral differences or line shapes - in practice it is impossible to visually evaluate the claim of a “distinct” spectrum for the mite exoskeleton. I recommend replacing/redoing Figure 2 at high resolution and supplementing it as follows: include a standard reference skin spectrum (control) and a “skin + mite” spectrum in a single panel, label all curves, and add zoomed insets for the 1000-1200 cm⁻¹ and 1600-1700 cm⁻¹ ranges where differences are reported, so that amplitude and peak-shape differences are visually evident. The present curves all look very similar and do not convincingly show visual differences even if the statistics are positive.
Third, Figure 5 is missing from the submitted file (the caption is present but the figure itself is absent). Because the text reports numerical values and ANOVA analysis for the 1072 cm⁻¹ band (and uses these results to argue diagnostic specificity), the corresponding graphical representation (boxplot) must be provided, with the number of measurements/samples and indication of statistical comparisons.
In addition, the following relatively straightforward corrections and clarifications are required: replace occurrences of “Ftir” with the standard capitalization “FTIR” (particularly in Sections 2.1 and 4.5).
Finally, a few optional suggestions to improve reproducibility: add raw, unprocessed spectra and/or a table of individual 1072 cm⁻¹ absorbance values for each sample in the Supplement; give more detailed clustering parameters and the rationale for the chosen cluster number (k-means initialization, number of iterations, distance metric); consider a brief comparison of potential spectral overlap with other ectoparasites , which the authors mention but do not substantiate with comparative data.
The study is interesting and has significant potential, but in its current form requires major revision, primarily to ensure adequate quality and completeness of the visual material (Figures 2 and 5) and to clarify the invasive/non-invasive terminology. After the requested corrections and provision of missing figures and raw data, the manuscript could be recommended for publication.
Author Response
Reviewer 3
The manuscript merits attention: the topic is timely, the approach is diagnostically interesting, the materials and methods are described in detail, regulatory and ethical aspects are documented, and the reported findings (identification of a chitin-specific band at 1072 cm⁻¹ and demonstration of its reproducibility across samples) have practical value as a complement to conventional histology. The authors performed careful acquisition and processing of FTIR data and applied appropriate multivariate analyses for tissue segmentation and parasite structure identification.
Nevertheless, I consider that in its present form the manuscript requires a major revision before it can be recommended for publication. The principal reasons are serious concerns about the clarity and verifiability of key illustrations and about the interpretation of one fundamental statement in the text.
First, the Introduction claims the absence of a “gold standard for non-invasive analysis” of mites. This formulation is misleading because the method used by the authors (FTIR on FFPE sections) requires biopsy sampling and therefore should be classified as an invasive diagnostic approach (minimally invasive from a clinical perspective, yes, but invasive in the sense that tissue must be removed). I ask the authors to explicitly state in what sense they use the terms “invasive / non-invasive” with respect to FTIR and related methods, and to clearly distinguish the role of FTIR as an adjunct to histological analysis from truly in-vivo, non-biopsy techniques (dermatoscopy, video dermoscopy, OCT, etc.).
Second, the Figure 2 image does not allow the reader to inspect real spectral differences or line shapes - in practice it is impossible to visually evaluate the claim of a “distinct” spectrum for the mite exoskeleton. I recommend replacing/redoing Figure 2 at high resolution and supplementing it as follows: include a standard reference skin spectrum (control) and a “skin + mite” spectrum in a single panel, label all curves, and add zoomed insets for the 1000-1200 cm⁻¹ and 1600-1700 cm⁻¹ ranges where differences are reported, so that amplitude and peak-shape differences are visually evident. The present curves all look very similar and do not convincingly show visual differences even if the statistics are positive.
Third, Figure 5 is missing from the submitted file (the caption is present but the figure itself is absent). Because the text reports numerical values and ANOVA analysis for the 1072 cm⁻¹ band (and uses these results to argue diagnostic specificity), the corresponding graphical representation (boxplot) must be provided, with the number of measurements/samples and indication of statistical comparisons.
In addition, the following relatively straightforward corrections and clarifications are required: replace occurrences of “Ftir” with the standard capitalization “FTIR” (particularly in Sections 2.1 and 4.5).
Finally, a few optional suggestions to improve reproducibility: add raw, unprocessed spectra and/or a table of individual 1072 cm⁻¹ absorbance values for each sample in the Supplement; give more detailed clustering parameters and the rationale for the chosen cluster number (k-means initialization, number of iterations, distance metric); consider a brief comparison of potential spectral overlap with other ectoparasites , which the authors mention but do not substantiate with comparative data.
The study is interesting and has significant potential, but in its current form requires major revision, primarily to ensure adequate quality and completeness of the visual material (Figures 2 and 5) and to clarify the invasive/non-invasive terminology. After the requested corrections and provision of missing figures and raw data, the manuscript could be recommended for publication.
AW: We sincerely thank the reviewer for the very positive overall assessment and the detailed, constructive comments. We have substantially revised the manuscript to address the issues raised regarding (i) the invasive/non-invasive terminology, (ii) the clarity and verifiability of Figures 2 and 5, (iii) minor technical points, and (iv) optional suggestions to improve reproducibility.
Invasive vs. non-invasive terminology and role of FTIR
We fully agree that our wording needed to be more precise. In the revised Introduction and Discussion, we now explicitly distinguish between:
Non-invasive, in-vivo bedside techniques (e.g., dermoscopy, video dermoscopy, OCT) applied directly to the patient, and
Ex vivo, biopsy-based, non-destructive methods such as FTIR microscopy applied to FFPE sections.
We have removed the ambiguous phrase “non-invasive nature of IR microscopy” and now describe FTIR as a label-free, non-destructive technique at the tissue level that requires an invasive biopsy and functions as an adjunct to histopathology, not as a non-invasive diagnostic test. We also added an explicit sentence clarifying in what sense we use the terms “invasive” and “non-invasive” in this manuscript (see revised Introduction and Discussion).
Figure 2 – clarity, resolution and visualisation of spectral differences
We agree that the original Figure 2 did not allow the reader to adequately inspect spectral line shapes and peak positions, particularly in the 1000–1200 cm⁻¹ and 1600–1700 cm⁻¹ regions. In the revised manuscript:
Figure 2 has been re-exported at high resolution with increased line thickness and substantially larger axis and legend fonts.
We have added two zoomed insets focusing on (i) the 1000–1200 cm⁻¹ carbohydrate region and (ii) the 1600–1700 cm⁻¹ Amide I region, so that amplitude and line-shape differences between mite exoskeleton, stratum corneum and dermis are visually evident.
All curves are clearly labelled, and the figure caption now explains the panel layout and highlights the 1072 cm⁻¹ band.
Because our dataset does not include independently acquired spectra from healthy control skin, we did not add an external “standard skin” spectrum; instead, we emphasise that dermis and stratum corneum from the same sections serve as internal reference compartments rather than true negative controls (as clarified in the Results and Limitations).
Figure 5 – missing figure
We apologise for the omission of Figure 5 in the original submission. We have now included the complete Figure 5 file in the revised manuscript.
Capitalisation of FTIR
We thank the reviewer for noting this. All occurrences of “Ftir” in section headings and text (e.g., Sections 2.1, 4.3 and 4.5) have been corrected to the standard “FTIR”.
Optional suggestions for reproducibility
We appreciate the reviewer’s suggestions and have implemented them as far as our dataset allows:
We now provide a table of individual 1072 cm⁻¹ absorbance values in the Supplementary Material (Supplementary Table S1) and refer to these in the Results and Data Availability Statement.
Section 4.5 (FTIR Imaging Analysis) and Section 2.2 (MIA) have been expanded to include more detailed clustering parameters (pre-processing, number of clusters, distance metric, rationale for k = 6, qualitative robustness criterion).
In the Discussion, we now explicitly address potential spectral overlap with other ectoparasites, emphasising that chitin is a conserved arthropod polysaccharide, that our dataset includes only Sarcoptes scabiei, and that species-level discrimination from other mites or insects cannot be claimed at this stage.
We hope that these revisions address the reviewer’s concerns regarding terminology, figure quality and completeness, and reproducibility. We are grateful for the reviewer’s positive assessment of the study’s potential and believe that the revised version more clearly reflects the preliminary, proof-of-concept nature of our work and its role as an adjunct to conventional histology.
Round 2
Reviewer 2 Report
Comments and Suggestions for Authors
The authors have addressed all the major issues. I recommend accepting the article for publication once the issue regarding figure formatting has been addressed. Figure 2 contains very small font size in the axis labels and in the figure itself.
Author Response
We thank the reviewer for this helpful remark. In the revised version of the manuscript, we have reformatted Figure 2 to comply with the MDPI figure guidelines. The updated version has now been included as the new Figure 2.